# Transcriptome Analysis of the Seed Shattering Mechanism in *Psathyrostachys juncea* Using Full-Length Transcriptome Sequencing

**DOI:** 10.3390/plants13243474

**Published:** 2024-12-11

**Authors:** Yuru Lv, Lan Yun, Xiaodi Jia, Yixin Mu, Zhen Li

**Affiliations:** College of Grassland Science, Inner Mongolia Agricultural University, Hohhot 010018, China; yurulv@163.com (Y.L.); jiaixaodi@163.com (X.J.); muyixin@emails.imau.edu.cn (Y.M.); lizhen0471@163.com (Z.L.)

**Keywords:** *Psathyrostachys juncea*, seed shattering, full-length transcriptome, synthesis of cellulose

## Abstract

Seed shattering (SS) functions are a survival mechanism in plants, enabling them to withstand adverse environmental conditions and ensure reproduction. However, this trait limits seed yield. *Psathyrostachys juncea*, a perennial forage grass with many favorable traits, is constrained by SS, limiting its broader application. To investigate the mechanisms underlying SS, second-generation Illumina sequencing and third-generation PacBio sequencing were conducted on abscission zone tissues of *P. juncea* at 7, 14, 21, and 28 days after heading. GO enrichment analysis identified several significant biological processes, including the “cell wall macromolecule catabolic process”, “cell wall polysaccharide catabolic process”, “hemicellulose catabolic process”, and “xylan catabolic process”, all involved in cell wall degradation. KEGG enrichment analysis showed that differentially expressed genes were predominantly enriched in pathways related to “starch and sucrose metabolism”, “fructose and mannose metabolism”, “phenylpropanoid biosynthesis”, “pentose and glucuronate interconversions”, and “galactose metabolism”, each linked to both the synthesis and degradation of the cell wall. Further analysis of the “starch and sucrose metabolism” pathway revealed genes encoding fructokinase, hexokinase, β-glucosidase, sucrose phosphate synthase, sucrose synthase, and endoglucanase, all of which affected cellulose content. Reduced cellulose content can alter cell wall structure, leading to SS. These findings provide new insights into the regulation of SS in *P. juncea* and offer valuable references for other species within the Poaceae family.

## 1. Introduction

Perennial grasses in the Poaceae family frequently display some degree of seed shattering (SS), a natural adaptive mechanism that evolved to help them withstand harsh environmental conditions and ensure reproduction [1]. However, SS significantly reduces seed yield and raises production costs, presenting a major challenge for forage seed production. Historically, breeding efforts for forage grasses have prioritized dry matter yield, nutritional quality, and tolerance to biotic and abiotic stresses, with relatively few studies addressing SS [2]. As a result, reducing SS and improving seed yield have become key goals in the domestication and breeding of forage grasses [3]. Understanding the mechanisms of SS has important theoretical implications for the genetic improvement of both wild and cultivated forage grasses.

The mechanisms of SS in herbaceous plants are categorized into two types: floret abscission, dominated by the pedicel, and spikelet abscission, controlled by the rachis [4,5,6]. *Elymus sibiricus*, *Elymus nutans*, and *Leymus chinensis* exhibit floret abscission [7]. *Bromus cartharticus* and *Panicum coloratum* undergo spikelet abscission [8]. The position of SS is known as the abscission zone, a specialized structure involved in the detachment of plant organs from the plant body [9]. During organ abscission, the abscission zone forms one or more layers of parenchyma cells, creating the abscission layer [10]. Plant organ separation is facilitated by the degradation of the middle lamella and softening of the cell walls [11,12].

*Psathyrostachys juncea* is a diploid (2n = 2x = 14) species of the Triticeae tribe in the Poaceae family. It is a cross-pollinating perennial grass, native to Central and Northern Asia, and has spread to Europe and North America through introduction and domestication [13]. This species exhibits early regrowth and a long green period, and retains high nutritional value in autumn. It is a typical cool season grass with strong resistance to wind, drought, salt, alkali, and grazing [14,15]. Its capacity for persistent use once established has made it a significant grazing grass and an essential species for ecological restoration in Northern China. Overall, *P. juncea* has considerable value for forage, breeding research, and economic applications. However, spikelet abscission at maturity leads to seed loss, limiting its practical application [13]. Up until now, research on SS in forage grasses are limited. Transcriptome sequencing analysis of abscission zone tissues in *Lolium perenne* revealed that *LpSH1* shows significantly higher expression level during floret development, indicating its role in regulating SS [16]. Similarly, transcriptome sequencing of *E. sibiricus* abscission zone tissues at various developmental stages indicated that enzymes related to cell wall degradation significantly contribute to SS [17]. Transcriptome analysis of *Stylosanthes* abscission zone tissues identified differentially expressed genes involved in lignin biosynthesis, cellulase synthesis, and plant hormone signal transduction [18]. Research on *Vicia sativa* revealed that β-glucosidase and endo-polygalacturonase cause cell wall degradation, leading to pod shattering [19]. Genome-wide association studies of *Setaria italica* identified previously unknown QTLs for a shattering-related locus, *Les1* [20]. No studies on SS in *P. juncea* have been reported. Transcriptome sequencing of abscission zone tissues in *P. juncea* at 7, 14, 21, and 28 days after heading was conducted to analyze the regulatory mechanisms of SS at the transcriptional level. These findings provide a reference for the improvement of SS traits and functional gene validation in *P. juncea* and other perennial grasses.

## 2. Results

### 2.1. Sequencing Quality

Transcriptome sequencing was performed on 12 samples of *P. juncea*, generating raw data ranging from 42,585,666 to 63,399,806 reads per sample. After quality control, clean data ranged from 41,956,422 to 62,547,880 reads, representing more than 98% of the raw data. Q20 bases accounted for more than 96.88%, Q30 bases exceeded 94.46%, and GC content ranged from 52.66% to 55.40% (Appendix A). These results suggest high sequencing quality, supporting further analysis.

### 2.2. Sample Analysis

Principal Component Analysis (PCA) can identify outlier samples as well as highly similar ones, providing direct evidence for the exclusion of outliers. As shown in Figure 1a, the 12 *P. juncea* samples were divided into four distinct clusters, with replicates from each time point clustered together. The transcriptome sequencing of *P. juncea* showed high reproducibility, and no outliers were identified. Therefore, all samples were included in the subsequent analysis. H7 was used as the control in these analyses. The number of differentially expressed genes (DEGs) increased over time, for both upregulated and downregulated genes (Figure 1b). The comparison “H28 vs. H7” revealed the most significant changes in gene expression, with 7886 genes upregulated and 3624 downregulated.

### 2.3. GO and KEGG Enrichment Analysis

A total of 2270 shared genes were identified across the three gene sets: H14 vs. H7, H21 vs. H7, and H28 vs. H7 (Figure 2). Enrichment analyses for GO and KEGG pathways were conducted on the shared DEGs (Figure 3). Significant biological processes in the GO enrichment analysis included the “cell wall macromolecule catabolic process”, “cell wall polysaccharide catabolic process”, “hemicellulose catabolic process”, and “xylan catabolic process”. Additionally, “response to monosaccharide”, “response to hexose”, and “auxin homeostasis” were also observed. Molecular functions enriched included “indole-3-acetic acid amido synthetase activity”, “amylase activity”, and “amylopectin maltohydrolase activity” (Figure 3a). Detailed information on the GO enrichment analysis is provided in Appendix A, and these processes are closely related to the SS of *P. juncea*. DEGs related to SS were enriched in the KEGG pathways, including “cutin, suberine, and wax biosynthesis”, “starch and sucrose metabolism”, “pentose and glucuronate interconversions”, “phenylpropanoid biosynthesis”, and “amino sugar and nucleotide sugar metabolism” (Figure 3b).

### 2.4. Short Time-Series Expression Miner Analysis (STEM)

STEM analysis was conducted to evaluate gene expression changes over time in identical sample types. A total of 2270 DEGs were categorized into 20 distinct expression profiles (0 to 19) (Figure 4). Based on expression patterns, these genes were further grouped into four clusters (0, 1, 2, and −1). The first three clusters displayed significant expression trends, while no significant trends were observed in cluster −1. GO enrichment analysis was performed for one profile from each of the clusters with significant trends (Figure 5; detailed gene information in Appendix A). In profile 19, gene expression consistently increased over time, with annotated genes such as “acetylserotonin O-methyltransferase”, “aldose reductase”, “indole-2-monooxygenase”, “transcription factor TGA”, and “xylan glycosyltransferase MUCI”. In contrast, genes in profile 0 demonstrated a consistent decrease in expression, including those annotated as “beta-amylase”, “endoglucanase”, “cellulose synthase”, “MADS-box transcription factor”, and “transcription factor MYB”. In profile 18, gene expression initially increased and subsequently declined, with annotated genes including “gibberellin 2-beta-dioxygenase”, “wall-associated receptor kinase”, “bZIP transcription factor”, “nuclear transcription factor”, and “transcription factor ICE”.

### 2.5. Gene Set Enrichment Analysis (GSEA)

GSEA was conducted using predefined gene sets, where genes were ranked according to their differential expression levels between two sample types. The analysis evaluated whether these predefined gene sets were enriched at either the top or bottom of the ranked list. The gene set with the highest number of DEGs, “H28 vs. H7” (5046 genes), was selected for this analysis (Figure 6). In H7 (Figure 6a), the significantly enriched KEGG pathways included “starch and sucrose metabolism”, “amino sugar and nucleotide sugar metabolism”, and “fructose and mannose metabolism”. Most genes within these pathways were upregulated in H7 and downregulated in H28. In contrast, the important pathways enriched in H28 (Figure 6b) included “phenylpropanoid biosynthesis” and “cutin, suberine and wax biosynthesis”, with certain genes upregulated in H28 and downregulated in H7. A few genes were downregulated in both H28 and H7. Detailed information about these pathways was provided in Appendix A. These gene sets were found to have significant biological relevance, based on NES (normalized enrichment score), *p*-value, and *p* adjust.

### 2.6. Weighted Gene Co-Expression Network Analysis (WGCNA)

#### 2.6.1. Identification of Key Module

The data were pre-filtered, after which 1718 DEGs were selected for WGCNA analysis. Through clustering, six distinct co-expression modules were identified. Significant upregulation and downregulation of genes were observed in each module across different samples (Figure 7a). The turquoise module exhibited the highest correlation with H7 (0.753) and H28 (−0.753) based on correlation coefficients and *p*-values for the modules and samples (Figure 7b), indicating it warrants further investigation.

#### 2.6.2. Enrichment Analysis of Module

KEGG enrichment analysis was conducted for the turquoise module (Figure 8), revealing significantly enriched pathways including “starch and sucrose metabolism”, “fructose and mannose metabolism”, “pentose and glucuronate interconversions”, “galactose metabolism”, “plant hormone signal transduction”, and “phenylpropanoid biosynthesis”. The highest-ranked pathway, “starch and sucrose metabolism”, was chosen for further analysis. Eight genes encoding enzymes in this pathway exhibited differential expression across developmental stages of *P. juncea* (Figure 9). The *INV* gene encodes β-fructofuranosidase, the *FRK* gene encodes fructokinase, the *BG* gene encodes β-glucosidase, the *SPS* gene encodes sucrose phosphate synthase, and the *PGI* gene encodes phosphoglucose isomerase. These five genes were highly upregulated in H7 and strongly downregulated in H28. The *HK* gene, encoding hexokinase, was upregulated in H7 and downregulated in H14, H21, and H28. The *SUS* gene, encoding sucrose synthase, and the *EG* gene, encoding endoglucanase, were upregulated in H7 and H14, and downregulated significantly in H28.

### 2.7. Quantitative Real-Time PCR (qRT-PCR) Validation of RNA-seq Data

To ensure the reliability of the transcriptome data, 10 genes related to SS in *P. juncea* were chosen for qRT-PCR validation (Figure 10). The results showed that the expression patterns of these genes in the abscission zone tissues during four developmental stages were consistent with the transcriptome sequencing results. Therefore, the reliability of the transcriptome sequencing data was confirmed.

## 3. Discussion

### 3.1. Application of Transcriptome Sequencing Technology

Second-generation transcriptome sequencing technologies have been extensively studied and applied in various plants, including *V. sativa* [21], *E. sibiricus* [22], *Eleusine coracana* [23], and *Triticum aestivum* [24]. According to published articles, third-generation full-length transcriptome sequencing technologies have been more frequently used in model plants and crops. In plant species with published reference genomes, comparing and analyzing third-generation transcriptome sequencing data with reference genome data has led to the discovery of many new genes and improved gene function annotation, as seen in wheat [25], sorghum [26], maize [27], and rice [28]. In plant species lacking genome sequence information, such as *Astragalus membranaceus* [29], *Cynodon dactylon* [30], and *Medicago sativa* [31], third-generation sequencing technology has been used to establish full-length transcript datasets at the transcriptome level. The PISO (Plant ISOform sequencing database), built using full-length transcript data from 19 published plant studies, serves as a valuable platform for integrative multi-omics analyses in plants [32]. By combining second- and third-generation transcriptome sequencing technologies, the long-read benefits of third-generation sequencing are used to obtain a large number of full-length transcripts, while the high accuracy of second-generation sequencing corrects the third-generation data for more reliable results. This method has been applied in *Drynaria roosii* [33], *Arabidopsis pumila* [34], and *Brassica alboglabra* [35], etc. In this study, for the same reason, we used the combined second- and third-generation transcriptome sequencing to obtain more complete transcript information from the abscission zone tissues of *P. juncea*.

### 3.2. Metabolic Pathways Related to SS

Several reports have described the role of plant hormone signal transduction and phenylpropanoid biosynthesis in SS pathways. Transcriptome analysis of young panicles from wild-type rice and the ssh1 mutant revealed that DEGs were enriched in biological processes such as metabolic regulation, gene expression regulation, transcription factor activity, plant hormone signal transduction, and phenylpropanoid biosynthesis [36]. These pathways are crucial for regulating SS in rice. Transcriptome analysis of abscission and non-abscission zones in *Oryza sativa* cultivars W517 and DZ129 revealed that DEGs in W517 were significantly enriched in plant hormone signal transduction, while those in DZ129 were enriched in phenylpropanoid biosynthesis [37]. Transcriptome analysis of weedy and cultivated rice also demonstrated significant enrichment of DEGs in phenylpropanoid biosynthesis, identifying abscisic acid as a key factor in SS in weedy rice [38]. Transcriptome analysis of seed peduncle structure in *Pennisetum alopecuroides* at five developmental stages showed significant enrichment of DEGs in pathways including phenylpropanoid biosynthesis, amino sugar and nucleotide sugar metabolism, fructose and mannose metabolism, galactose metabolism, pentose and glucuronate interconversions, plant hormone signal transduction, and starch and sucrose metabolism [39]. KEGG enrichment analysis of abscission zone tissues from *P. juncea* at different developmental stages showed significant enrichment of DEGs in the pathways of starch and sucrose metabolism, pentose and glucuronate interconversions, phenylpropanoid biosynthesis, and amino sugar and nucleotide sugar metabolism. Further analysis of the turquoise module, identified through WGCNA, also showed enrichment of DEGs in the fructose and mannose metabolism, galactose metabolism, and plant hormone signal transduction pathways. Among them, starch and sucrose metabolism play a key role in regulating SS in *P. juncea*. It can be inferred that the key metabolic pathways that affect SS vary among different plants.

### 3.3. Analysis of Starch and Sucrose Metabolism

Sucrose is crucial in regulating plant growth, fruit ripening, and the synthesis of starch and cellulose. It acts as an essential form of carbohydrate storage and transport in plants. Sucrose is transported through the phloem to tissues needing carbon and energy, promoting cell growth [40]. Sucrose influences metabolism by regulating gene expression. Sucrose transport is associated with cellulose synthesis, a key component of the plant cell wall [41]. Therefore, studying how sucrose metabolism impacts cell wall structure is important. Current research identifies three main enzymes involved in sucrose metabolism: β-fructofuranosidase (INV), sucrose synthase (SUS), and sucrose phosphate synthase (SPS) [42,43]. SPS promotes sucrose synthesis and supports plant growth, cell differentiation, and fiber cell wall formation. Expression of the spinach *SPS* gene in cotton enhances sucrose synthesis, improving fiber length and quality [44]. In *Arabidopsis*, overexpression of the *SPS* gene in tobacco increased sucrose content and stem length, indicating a relationship between sucrose levels and fiber growth [45]. Similarly, sucrose accumulation in sugarcane stems is achieved by upregulating *SPS* gene expression and activity [46]. The *AcSPS5* gene in pineapple has been found to be primarily expressed in the peduncle and peel [47]. *SPS* gene expression was downregulated during the abscission zone development in *P. juncea*. The downregulation of *SPS* gene expression may regulate sucrose content, leading to reduced cellulose synthesis. As a result, cell wall stability was weakened, possibly triggering SS. SUS transports sucrose into multiple pathways, providing precursor substances such as UDP-glucose for the biosynthesis of cellulose and starch in the cell wall [48]. Previous studies have demonstrated the importance of SUS in cotton fiber development [49]. Overexpression of the *SUS* gene from potato tubers in cotton leads to longer fibers compared to non-transgenic plants [50]. In *P. juncea*, downregulation of the *SUS* gene was associated with reduced cellulose synthesis in the plant cell wall during seed maturation, leading to SS. INV, which hydrolyzes sucrose into glucose and fructose, plays a critical role in the cell wall. The *INV* gene promotes tomato fruit growth but inhibits seed development when its activity is elevated [51]. In *P. juncea*, decreased *INV* gene expression during abscission zone development altered the role of sucrose in the cell wall. Fructose and glucose, catalyzed by hexokinase (HK) and fructokinase (FRK), enter glycolysis, the TCA cycle, or other secondary metabolic pathways [52]. FRK, the key enzyme in fructose metabolism, is essential for growth and cellulose synthesis. Studies in *Arabidopsis* [53], potato [54], and poplar [55] indicate that suppression of *FRK* gene expression restricts plant growth and cellulose synthesis. When the *SlFRK3* and *SlFRK2* genes were simultaneously suppressed in tomatoes, plant growth was restricted, mature leaves wilted, and the cell walls thinned and collapsed [56]. Joint suppression of *SlFRK1* and *SlFRK2* results in reduced xylem areas, fewer vessels, and thinner phloem cell walls [57]. In *P. juncea*, downregulation of the *HK* and *FRK* genes indicated that decreased activity reduced cellulose synthesis in the abscission zone cell wall, leading to SS. Cellulose in plant cell walls primarily consists of β-linked glucosyl units. β-glucosidase (BG) maintains cell wall structure by degrading oligosaccharides and releasing lignin monomers from glycosides [58]. During seed development, BG hydrolyzes starch and cellulose, facilitating glucose accumulation, which supports cell wall synthesis [59]. In rice, the *Os3BGlu7*, *Os3BGlu8*, and *Os7BGlu26* genes hydrolyze cellulose oligosaccharides, releasing glucose to support cell wall development [60,61]. In *P. juncea*, downregulation of the *BG* gene during seed maturation indicated that reduced BG activity caused structural changes in the cell wall, leading to SS. Endoglucanase (EG), also termed cellulase, is part of the glycosyl hydrolase 9 (*GH9*) gene family. EG plays a role in several physiological processes, such as maintaining cell wall structure, tissue differentiation, fruit ripening, and organ abscission [62]. In *P. juncea*, the *EG* gene and the phosphoglucose isomerase (*PGI*) gene regulated cellulose synthesis in the cell wall, affecting SS.

The differentially expressed genes identified may play a role in the SS of *P. juncea*, though their specific functions still require further validation. This offers a valuable pool of candidate genes for further studies on the mechanisms underlying SS in *P. juncea*.

## 4. Materials and Methods

### 4.1. Plant Materials

The experiment was conducted in Hohhot, Inner Mongolia (40°44′ N, 111°50′ E, 1048 m altitude). Ten high SS *P. juncea* (H) plants were selected from a natural population for study based on prior evaluations of their SS rate (Appendix A). Samples were taken on days 7, 14, 21, and 28 after heading and labeled H7, H14, H21, and H28 [17]. Three biological replicates were obtained for the four samples. For each plant, spikelets were collected from the middle of the spike, including the junction between the spikelets and rachis, and the rachis itself. (The target area measured was about 1 mm in length. Due to the short rachilla and high SS of *P. juncea*, other parts were removed, leaving only the junction between spikelets and rachis, and the rachis itself.)

### 4.2. Transcriptome Sequencing

Equal amounts of samples from each plant at the same developmental stage were mixed and divided into three biological replicates. The samples were sent to Majorbio Bio-Pharm Technology Co., Ltd. (Shanghai, China) for RNA extraction and quality control. Samples that passed quality control were used for library construction and second- and third-generation sequencing. After mixing the RNA from different samples evenly, the SMARTer PCR cDNA Synthesis Kit (Clontech, Mountain View, CA, USA) was used to reverse transcribe the total RNA into cDNA, which was sequenced using a Sequel II sequencer (Pacbio, Menlo Park, CA, USA). Twelve cDNA libraries from various samples were sequenced using a NovaSeq X Plus sequencer (Illumina, San Diego, CA, USA).

### 4.3. Quality Control

Polymerase reads shorter than 50 bp, sequences with an accuracy below 0.90, and redundant sequences were filtered out from the third-generation sequencing data. Sequence completeness was assessed using BUSCO software (Version 3.0.2) [63], and full-length transcripts were obtained. Raw data from second-generation sequencing were quality controlled using fastp software (Version 0.23.4) [64]. Low-quality sequences shorter than 20 bp and adapter sequences were removed, resulting in clean data. Clean data were de novo assembled using Trinity software (Version v2.8.5) [65]. Assembly results were evaluated using TransRate software (Version v1.0.3) [66], and redundant sequences were removed with CD-HIT software (Version v4.5.7) [67]. Finally, sequence completeness was reassessed using BUSCO software. Transcript data from second- and third-generation sequencing were integrated using TAMA software (Version v1.0) [68]. Long-read information from third-generation data was used to optimize the assembly results of the second-generation data, generating more complete transcripts.

### 4.4. Differential Expression Analysis and Functional Enrichment

Gene abundances were quantified with the RSEM software (Version 1.3.1) [69] and normalized by the fragments per kilobase of transcript per million mapped reads (FPKM) method [70]. Differential expression analysis was performed with the DESeq2 software (Version 1.42.0) [71], and genes with fold change (FC) ≥ 1 and false discovery rate (FDR) < 0.05 were considered significantly DEGs. GO enrichment analysis was conducted with Goatools software, while KEGG enrichment analysis was performed using the Python scipy software (Version 1.14.0). Enrichment was deemed significant when the *p*-value was below 0.05.

### 4.5. Data Analysis

Time-series expression trend analysis using STEM [72] investigates gene expression patterns at multiple time points. Functional enrichment analysis of genes linked to specific expression patterns follows to explore their biological functions. In our study, gene expression changes across four developmental stages of the abscission zone in high SS *P. juncea* were analyzed. Subsequently, GO enrichment analysis was conducted to determine the functions of the STEM expression patterns.

GSEA [73] ranks genes by their differential expression in samples and tests for enrichment at either the top or bottom of the ranked list. Analyzing changes in the entire gene set allows detection of genes with subtle expression differences that may hold significant biological relevance. In our analysis, a threshold of *p*-value < 0.05, *p* adjust < 0.25, and |NES| > 1 was used to identify significant gene sets.

A total of 1718 DEGs with FPKM > 5 were selected for WGCNA [74] analysis. Data were preprocessed before analysis to ensure that the correlation strengths between genes conformed to a scale-free distribution. A soft threshold power of 14 was applied to construct the co-expression network. Dynamic tree cutting was applied for module clustering, and the correlation between modules and samples was expressed with *p*-values.

### 4.6. qRT-PCR Analysis

Ten DEGs related to SS in *P. juncea* were selected for validation. RNA was reverse transcribed into cDNA using the HiFiScript cDNA Synthesis Kit (Cowin Bio, Shanghai, China) for qRT-PCR validation. Actin was used as the reference gene [75]. Primers were designed using NCBI Primer-BLAST (Appendix A) and synthesized by Beijing Genomics institution (BGI, Beijing, China). The relative expression levels of the DEGs were calculated using the 2^−ΔΔCt^ method [76]. The qRT-PCR analysis was performed using three technical ones per biological replicate.

## 5. Conclusions

In summary, our findings elucidated that biological processes such as the “cell wall macromolecule catabolic process”, “cell wall polysaccharide catabolic process”, “hemicellulose catabolic process”, and “xylan catabolic process” were linked to cell wall degradation. Pathways such as “starch and sucrose metabolism”, “fructose and mannose metabolism”, “phenylpropanoid biosynthesis”, “pentose and glucuronate interconversions”, and “galactose metabolism” were involved in both cell wall synthesis and degradation. In the “starch and sucrose metabolism” pathway shown in Figure 9, genes encoding fructokinase, hexokinase, β-glucosidase, sucrose phosphate synthase, sucrose synthase, endoglucanase, β-fructofuranosidase, and phosphoglucose isomerase were downregulated during the development of the abscission zone in *P. juncea*. This downregulation impacted cellulose synthesis, ultimately leading to changes in cell wall structure, resulting in SS. These findings deepened our understanding of the mechanisms underlying SS in *P. juncea*.

## Figures and Tables

**Figure 1 plants-13-03474-f001:**
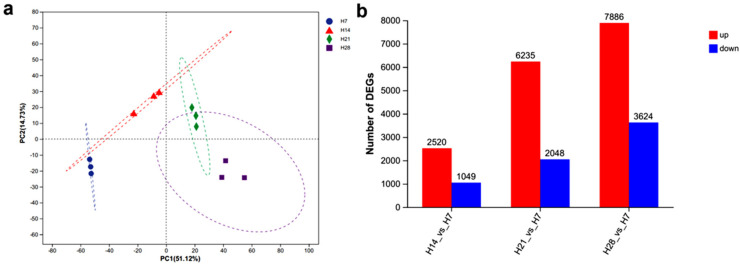
(**a**) The PCA of different *P. juncea* samples. (**b**) Statistics of differentially expressed genes in the H14 vs. H7, H21 vs. H7, and H28 vs. H7 comparisons.

**Figure 2 plants-13-03474-f002:**
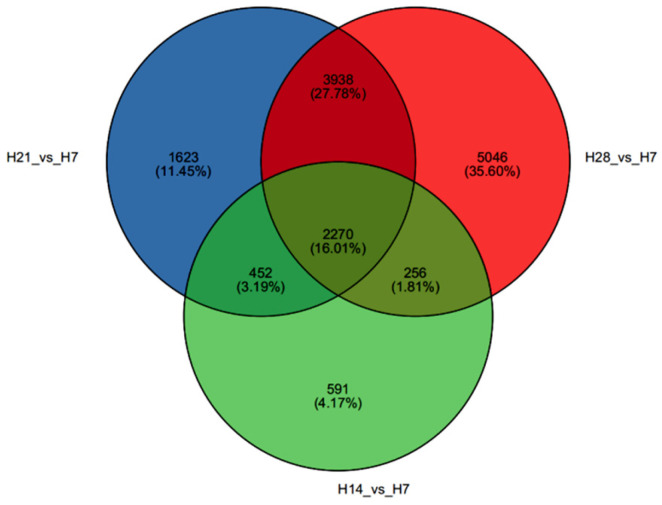
Venn diagram of samples from different developmental stages of *P. juncea*.

**Figure 3 plants-13-03474-f003:**
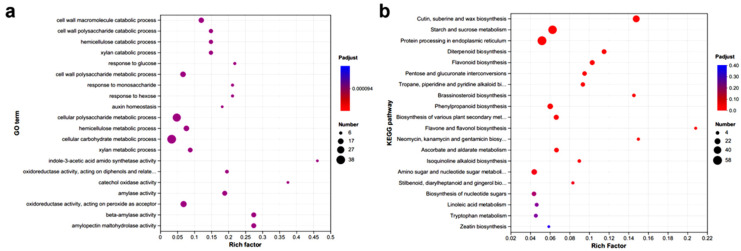
(**a**) GO and (**b**) KEGG enrichment analyses of shared genes at different developmental stages in *P. juncea*.

**Figure 4 plants-13-03474-f004:**
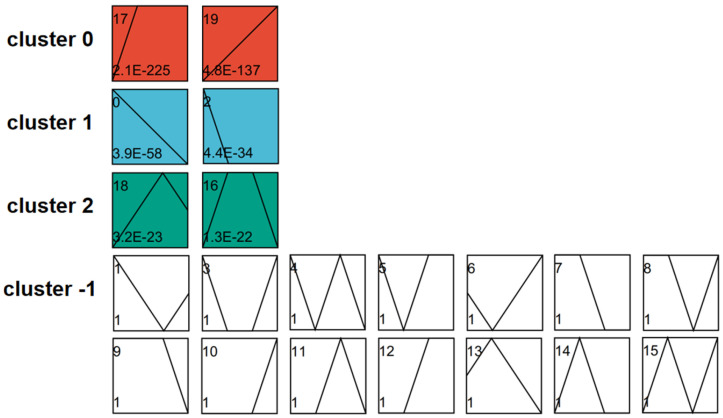
STEM analysis of shared genes at different developmental stages in *P. juncea.* Each rectangle corresponds to a profile. The number in the upper left corner represents the profile number, while the straight line indicates the trend of gene expression changes over time. The significance level (*p*-value) is shown in the lower left corner. Profiles with identical colors belong to the same cluster, indicating that they exhibit similar trends. Profiles without color exhibit no significant changes in their time-series patterns.

**Figure 5 plants-13-03474-f005:**
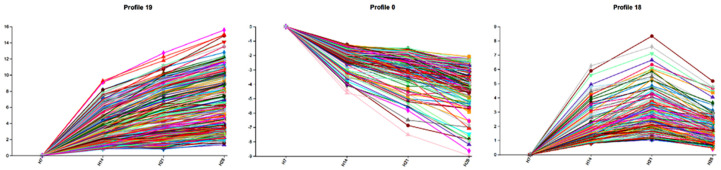
Profile trend chart derived from STEM analysis. Each line represents a gene.

**Figure 6 plants-13-03474-f006:**
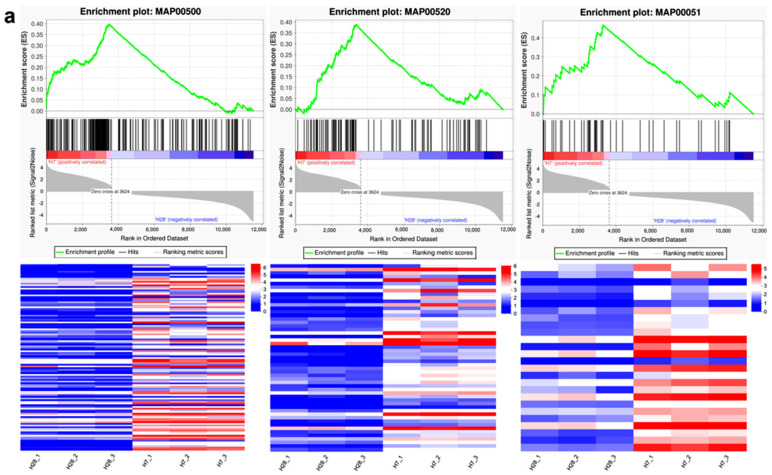
The significantly enriched pathways in (**a**) H7 and (**b**) H28 are presented. The green line indicates the enrichment of gene set members at the extremes of the ranked list, as measured by the enrichment score (ES). Positive values in the gray region suggest gene association with the left sample, whereas negative values suggest association with the right sample. Black vertical lines denote the occurrence of genes within this gene set.

**Figure 7 plants-13-03474-f007:**
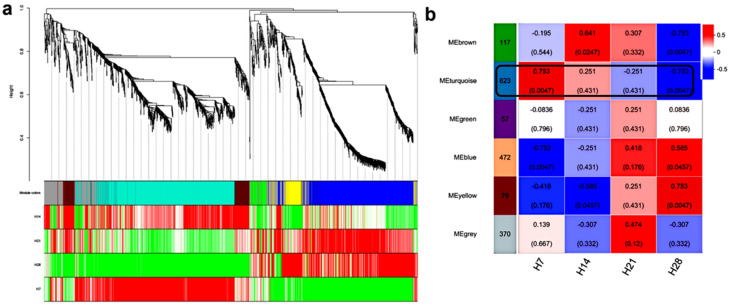
(**a**) A hierarchical cluster tree was generated for the identification of co-expression modules using WGCNA. Each leaf on the tree represents a single gene. The tree consists of six modules, each represented by a distinct color. Red denotes upregulated genes, and green denotes downregulated genes. (**b**) A heat map was generated to display correlations between modules and various samples. Each row represents a module, and each column corresponds to a sample. Red cells indicate positive correlations, while blue cells represent negative correlations. The number at the top of each cell shows the correlation coefficient, while the number in parentheses below denotes the *p*-value.

**Figure 8 plants-13-03474-f008:**
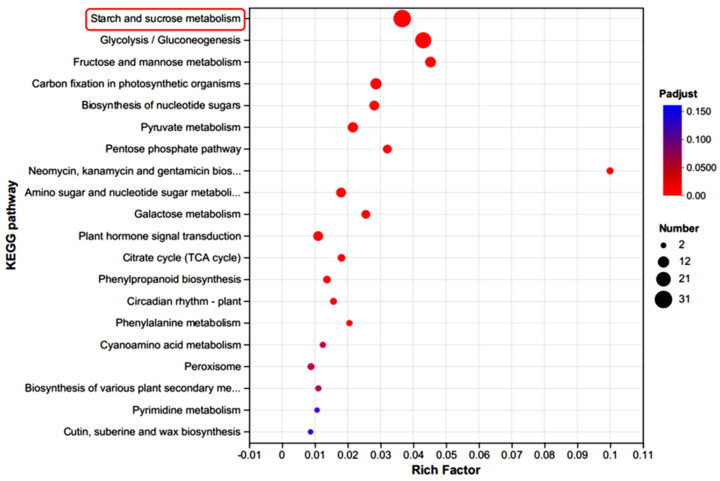
KEGG enrichment analysis was conducted on the turquoise module.

**Figure 9 plants-13-03474-f009:**
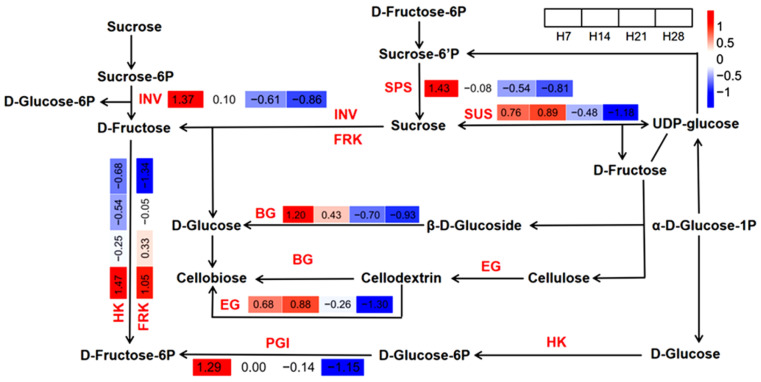
The starch and sucrose metabolism (map00500) pathway in *P. juncea* was identified. Color represents gene expression levels in the samples, where deeper colors denote higher expression levels.

**Figure 10 plants-13-03474-f010:**
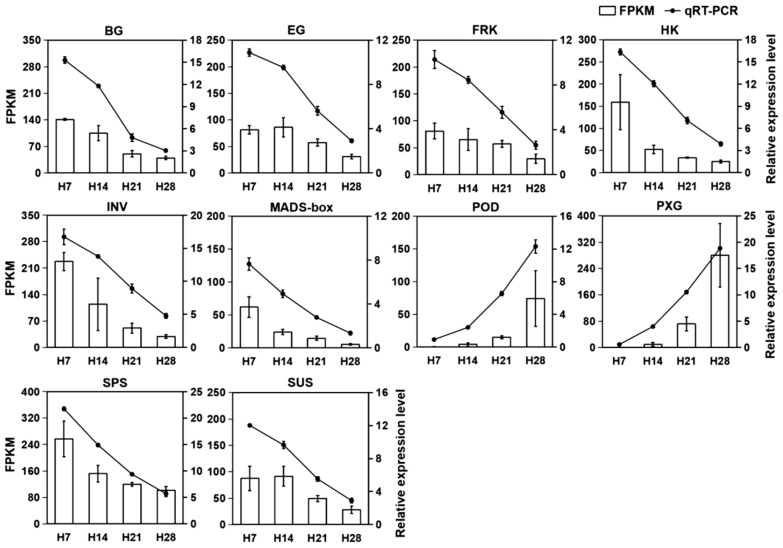
Expression of seed shattering unigenes in *P. juncea* quantified by RNA-Seq and qRT-PCR analysis. White bars represent changes in transcript abundance based on FPKM values from transcriptome sequencing (left *y*-axis). Lines indicate relative expression levels measured by qRT-PCR (right *y*-axis). Error bars represent the mean ± standard deviation. BG: β-glucosidase. EG: endoglucanase. FRK: fructokinase. HK: hexokinase. INV: β-fructofuranosidase. MADS-box: MADS-box transcription factor. POD: peroxidase. PXG: peroxygenase. SPS: phosphate synthase. SUS: sucrose synthase.

## Data Availability

Transcriptome data used in the research are available in the NCBI SRA database under accession numbers PRJNA1146567. The addresses are as follows: (https://www.ncbi.nlm.nih.gov/bioproject/PRJNA1146567 (accessed on 10 August 2024)).

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
