# Peer review of "Transcriptome Analysis of the Seed Shattering Mechanism in Psathyrostachys juncea Using Full-Length Transcriptome Sequencing"

_plants, 2024, doi:10.3390/plants13243474_

Round 1

Reviewer 1 Report

Comments and Suggestions for Authors

Lv and colleagues conducted a transcriptome analysis on the abscission zone tissues of Psathyrostachys juncea. Their findings suggest that biological processes related to cell wall synthesis and degradation may contribute to seed shattering, a significant factor limiting seed yield. The study was well-designed, the data are robust and well-presented, and the manuscript is thoroughly prepared. I am pleased to recommend its acceptance for publication, provided the authors address the following minor points:

1. It is suggested to include a figure illustrating the morphological differences in the abscission zone tissues at 7, 14, 21, and 28 days after heading. 

2. In the DEG, GO, and KEGG analyses, why are comparisons only made against H7? Comparisons between other days after heading, particularly between neighboring time points, would provide a more comprehensive understanding of temporal changes. 

3. In lines 162 and 364, the number "1718" should be formatted as "1,718." 

Author Response

Dear revievers,

We sincerely thank the editor and all reviewers for your valuable feedback and for giving us the opportunity to re-submit a revised manuscript. The certainly help us to improve the quality of our manuscript. We considered comments from the reviewer and made a number of revisions for the manuscript. All comments are addressed on a point-by-point basis below. We have re-uploaded the revised manuscript.

Best regards!

Yours sincerely,

Yuru Lv, Lan Yun, Miaomiao Jia, Yixin Mu, Zhiqiang Zhang

Comments 1: It is suggested to include a figure illustrating the morphological differences in the abscission zone tissues at 7, 14, 21, and 28 days after heading. 

Response 1: We fully agree with the reviewers’ suggestions. Our comprehensive study focuses on the abscission zone (AZ) at the connection between the spikelet and the rachis in P. juncea. We conducted morphological, histological, physiological, and transcriptomic analyses of the AZ across four developmental stages in both high-seed-shattering and low-seed-shattering P. juncea. This work has been accepted by BMC Plant Biology. The current study emphasizes transcriptomic analysis of the AZ at different developmental stages in high-seed-shattering P. juncea. Including morphological images of the AZ at different developmental stages would overlap with our previous publication. We kindly hope for your understanding on this matter.

Comments 2: In the DEG, GO, and KEGG analyses, why are comparisons only made against H7? Comparisons between other days after heading, particularly between neighboring time points, would provide a more comprehensive understanding of temporal changes. 

Response 2: We appreciate your insightful comments. In Section 2.2, it has already been stated that H7 serves as the control group. This study focuses on the comparative analysis between the treatment groups (H14, H21, H28) and the control group.

Comments 3: In lines 162 and 364, the number "1718" should be formatted as "1,718." 

Response 3: We sincerely thank you for your meticulous review. We have carefully revised the manuscript in accordance with your valuable suggestions.

Reviewer 2 Report

Comments and Suggestions for Authors

The manuscript fits into the general current trend, based on the transcriptome analysis, in this case of Psathyrostachys juncea, at several points in ontogeny to detail the seed-shattering process.

However, I have a few comments. My most important point is that I need the research to be purposeful. The introduction and discussion should emphasise the seed-shattering process, emphasising its importance in the ontogeny of grasses and its significance for humans.

The descriptions of the resulting part are significantly lacking in the description of the samples, both in the text and in the descriptions of the figures. The figures should be self-explanatory, but they are not. Please expand the description of each figure. I don't know what H is, how many biological repetitions there were, how many technical ones, or what the statistics shown are.

The description under Figure 10. should include expansions of the gene name abbreviations.

Author Response

Dear revievers,

We sincerely thank the editor and all reviewers for your valuable feedback and for giving us the opportunity to re-submit a revised manuscript. The certainly help us to improve the quality of our manuscript. We considered comments from the reviewer and made a number of revisions for the manuscript. All comments are addressed on a point-by-point basis below. We have re-uploaded the revised manuscript.

Best regards!

Yours sincerely,

Yuru Lv, Lan Yun, Miaomiao Jia, Yixin Mu, Zhiqiang Zhang

Comments 1: The introduction and discussion should emphasise the seed-shattering process, emphasising its importance in the ontogeny of grasses and its significance for humans.

Response 1: Thank you for your valuable feedback. This study discusses metabolic pathways related to seed shattering in the discussion section (3.2), with a particular emphasis on the "starch and sucrose metabolism" pathway. As highlighted in the introduction, understanding the mechanisms of seed shattering holds significant theoretical value for the genetic improvement of both wild and cultivated forage grasses. Our findings provide valuable references for improving seed shattering traits and validating functional genes in P. juncea and other perennial grasses.

Comments 2: The descriptions of the resulting part are significantly lacking in the description of the samples, both in the text and in the descriptions of the figures. The figures should be self-explanatory, but they are not. Please expand the description of each figure.

Response 2: Thank you for your comment. We have referred to articles previously published in this journal to draft the results section and have reviewed it again to ensure that the figure interpretations remain objective, concise, and clear.

Comments 3: I don't know what H is, how many biological repetitions there were, how many technical ones, or what the statistics shown are.

Response 3: Thank you for your suggestion. In section 4.1, we specified that "H" represents high seed shattering P. juncea and added "three biological replicates were obtained for the four samples." In section 4.6, we included "the qRT-PCR analysis was performed using three technical replicates per biological replicate."

Comments 4: The description under Figure 10. should include expansions of the gene name abbreviations.

Response 4: Thank you for your suggestion. We have included the full names of all genes in the legend of Figure 10 for better clarity and understanding.
